# Experimental Water Activity Suppression and Numerical Simulation of Shale Pore Blocking

Yansheng Shan [1,2], Hongbo Zhao [1,2,*], Weibin Liu [1,*], Juan Li [1,2], Huanpeng Chi [1,2], Zongan Xue [1], Yunxiao Zhang [1,2] and Xianglong Meng [1,2]

1 Oil and Gas Survey, China Geological Survey, Beijing 100083, China; danyansheng@mail.cgs.gov.cn (Y.S.); rosejuanli@126.com (J.L.); chp2121@126.com (H.C.); xuezongan@126.com (Z.X.); zhangyunxiao@mail.cgs.gov.cn (Y.Z.); mengxianglong001@mail.cgs.gov.cn (X.M.)
2 Engineering Research Center of Unconventional Oil and Gas, China Geological Survey, Beijing 100037, China
* Correspondence: zhongbo@mail.cgs.gov.cn (H.Z.); liuweibin@mail.cgs.gov.cn (W.L.)

**Abstract:** The nanoscale pores in shale oil and gas are often filled with external nanomaterials to enhance wellbore stability and improve energy production. And there has been considerable research on discrete element blocking models and simulations related to nanoparticles. In this paper, the pressure transmission experimental platform is used to systematically study the influence law of different water activity salt solutions on shale permeability and borehole stability. In addition, the force model of the particles in the pore space is reconstructed to study the blocking law of the particle parameters and fluid physical properties on the shale pore space based on the discrete element hydrodynamic model. However, the migration and sealing patterns of nanomaterials in shale pores are unknown, as are the effects of changes in particle parameters on nanoscale sealing. The results show that: (1) The salt solution adopts a formate system, and the salt solution is most capable of blocking the pressure transmission in the shale pores when the water activity is 0.092. The drilling fluid does not easily penetrate into the shale pore space, and it is more capable of maintaining the stability of the shale wellbore. (2) For the physical blocking numerical simulation, the nanoparticle concentration is the most critical factor affecting the shale pore blocking efficiency. Particle size has a large impact on the blocking efficiency of shale pores. The particle diameter increases by 30% and the pore-blocking efficiency increases by 13% when the maximum particle size is smaller than the pore exit. (3) Particle density has a small effect on the final sealing effect of pore space. The pore-plugging efficiency is only increased by 4% as the particle density is increased by 60%. (4) Fluid viscosity has a significant effect on shale pore plugging. The increase in viscosity at a nanoparticle concentration of 1 wt% significantly improves the sealing effectiveness, specifically, the sealing efficiency of the 5 mPa-s nanoparticle solution is 16% higher than that of the 1 mPa-s nanoparticle solution. Finally, we present a technical basis for the selection of a water-based drilling fluid system for long horizontal shale gas drilling.

**Keywords:** shale; water activity; nanoscale pores; numerical simulation; salt ion inhibition



## 1. Introduction

In recent years, natural resources such as shale oil and gas have garnered significant attention from scholars due to the shifting dynamics of global energy resource consumption and the efficient development and utilization of unconventional resources [1–4]. Notably, China has reported proven recoverable reserves of shale gas amounting to $35 \times 10^8$ t since 2020 [5]. Enhancing the efficient exploitation of shale oil and gas has emerged as a crucial factor to ensure China's economic development. Horizontal well drilling technology is widely recognized as a key technique for enhancing shale oil and gas production. However, prolonged implementation of horizontal drilling may yield increased reservoir hydration, damage, and even instability and collapse of the borehole [6].

Shale formation is characterized by a significant abundance of micropores and reactive clay minerals such as kaolinite, saponite, and montmorillonite. There are two main reasons for wellbore destabilization. Firstly, the invasion of drilling fluid into the micropores prompts hydration and expansion of clay minerals. Secondly, reducing the differential pressure can result in wellbore destabilization because pressure transmission facilitates the injection of drilling fluid into the shale formation. The stability of the wellbore is greatly influenced by the properties of the drilling fluids. The flow of drilling fluid between fractures can interact with minerals, and reactants may precipitate, causing blockages or dissolve and creating new paths [7,8]. Consequently, inorganic salt and plugging agents are commonly employed to mitigate hydration and seal porous formations [9–12].

Plugging agents are primarily employed to address well leakage issues during the drilling and completion of shale gas horizontal wells, thereby ensuring the smooth production of shale gas. The compatibility between shale and water-based drilling fluids is of utmost importance, as it can give rise to challenges related to clay expansion and wellbore stability [13]. In recent years, domestic and international research efforts have focused on investigating the impact of various nanoparticle parameters on shale plugging efficacy. Nanoparticles have shown promise in effectively sealing the nanoscale micropores within shale formations, thus retarding shale hydration and expansion [14]. Sensoy initially proposed the incorporation of nanoparticles into water-based drilling fluids as a means to enhance the plugging of shale pore throats. Their study demonstrated that the appropriate application of nanoparticles with suitable sizes significantly reduced fluid penetration into shale formations [15]. Taraghikhah further investigated the plugging capability of silicon dioxide nanoparticles at low concentrations (not exceeding 1%) within shale pores. They employed scanning electron microscopy (SEM) imaging techniques to analyze the mechanisms of pore plugging. Visual observations revealed that silicon dioxide nanoparticles effectively covered the shale surface, thereby blocking the shale pores [16]. Yang [17] conducted numerical simulations and experimental verifications to identify the main factors influencing particle clogging, including particle concentration, particle size, and pore roughness. Furthermore, these parameters were examined to assess their impact on blocking efficiency under different fluid viscosities [17]. The experimental results demonstrated a positive correlation between particle size, concentration, and fluid viscosity with the plugging effect.

The hydration, swelling, and degradation of shale and clay are inherent phenomena, but they can be mitigated through the use of inorganic salts and inhibitors. Traditional experimental evaluation techniques aimed at assessing the macroscopic effects of inhibiting shale hydration face challenges in capturing microscopic mechanisms and processes involving inorganic salts. In recent years, numerous experiments conducted by domestic scholars have been dedicated to studying the microscopic mechanisms underlying the inhibition of shale hydration by inorganic salts. Chen developed a coupled hydrochemical–mechanical model for simulating water–rock interactions in fractured shale during post-fracturing flowback. The results show that the swelling volume of clay minerals occupies the pore space, leading to a decrease in matrix porosity, while mineral dissolution increases matrix porosity and solute concentration of the aqueous phase in the matrix pore space. Clay expansion mainly affected the shape of the porosity curve [18]. In a study conducted by Wang Yepeng [19], an investigation of water–shale interactions was carried out. The findings revealed that shales encompass various clay minerals, each exhibiting distinct hydration modes. Moreover, different inorganic salt solutions were found to exhibit varying degrees of efficacy in inhibiting the hydration of illite and Na-montmorillonite. Wang developed a characterization model and reservoir numerical simulation method capable of describing changes in reservoir porosity and permeability caused by salt dissolution and recrystallization. It was shown that salt dissolution increases reservoir porosity and permeability, and the magnitude of the change is proportional to the amount of water passing through the reservoir, which can improve the flow capacity of shale oil.

On the contrary, recrystallization leads to a decrease in reservoir porosity and permeability [20]. Yang Xianyu [21] proposed a dynamic model based on the osmotic pressure of the Longmaxi Formation shale. The study examined the influence of different substances, namely NaCl, KCl, CaCl$_2$, and HCOONa, on the osmotic pressure and the osmotic pressure law of the Longmaxi Formation shale. The findings indicate that the osmotic pressures of KCl and HCOONa decrease as salinity increases. In the case of NaCl, the osmotic pressure initially decreases and then increases with increasing salinity, reaching its minimum value at approximately 0.25 salinity of NaCl solution about the Longmaxi Formation shale. Yayun Zhang [22] conducted molecular dynamics simulation experiments to develop molecular dynamics models for four representative cations involved in inhibiting clay mineral hydration. The study comprehensively evaluated the microscopic kinetic mechanisms underlying the inhibition of clay mineral hydration by these typical inorganic cations. Additionally, the authors analyzed the variation patterns of cation hydration inhibition performance concerning temperature, pressure, and ion type. The results demonstrated that cations facilitate the contraction of interlayer spacing, compress fluid intrusion channels, diminish the intrusion capacity of water molecules, enhance negative charge balance ability, and reduce interlayer electrostatic repulsion force. Moreover, the study finds that cation inhibition of montmorillonite hydration weakens with increasing temperature, while the effect reverses with pressure.

Considering the economic cost and greenness, water-based drilling fluids, which are cheaper and almost harmless to the environment, are more promising for application compared to oil-based drilling fluids. At present, the sealing effects of nanoparticle drilling fluid on shale pore space are mostly limited to physical experimental data, and the transportation, dynamic accumulation, and microscopic sealing mechanisms of nanoparticles in drilling fluid after invading shale pore space are not clear. Therefore, nanoparticle sealing efficiency and real-time invasion volume of drilling fluid are not easy to determine quickly under different fluid physical characteristics and discrete element parameters. In this paper, a discrete element fluid–solid coupling model is provided, which is able to analyze and quantitatively evaluate the plugging performance of different plugging materials for shale pore space.

i    In this paper, we study the effect of different parameters of nanoparticles on the physical plugging performance of shale pores through hydrodynamic simulation.
ii   The effects of salt solution on the chemical percolation performance of shale under the effects of differential pressure are studied using a shale pressure transmission experimental apparatus.
iii  The effects of fluid viscosity on the blocking of shale pores are investigated using numerical simulations.
iv   The research content can provide a basis for inhibiting shale hydration and maintaining the stability of shale gas horizontal wells.

## 2. Experimental Materials and Methods

### 2.1. Experimental Materials

Nano silica dispersion, sodium chloride, potassium chloride, calcium chloride, sodium formate, and potassium formate were used. The experimental shale was taken from Xiushan County, south of Chongqing. Various shale cores (approximately 0.5 cm in height and 2.5 cm in diameter) were drilled using a core drilling machine for the pressure transmission experiment.

### 2.2. Experimental Instruments

An X-ray diffractometer (Malvern Panalytical, Amsterdam, Netherlands), scanning electron microscope (SEM) (FEI, hillsborough, NC, USA), PTE shale pressure transferring device (TC, HaiAn, China), Novasina Labswift water activity tester (Novasina, Zurich, Switzerland), QBZY automatic surface tension meter (Fangrui, Shanghai, China), core

cutter and press, core drilling machine, and LCMP-1A metallurgical sample grinding and polishing machine (TC, HaiAn, China) were used.

### 2.3. Shale X-ray Diffraction and Microstructure Analysis

Shale samples were collected from the Longmaxi Formation. An X-ray diffraction (XRD) analysis revealed that the shale from the Longmaxi Formation has approximately 58% quartz content and 25% clay mineral content (Table 1). Consequently, these shale samples exhibited a predominance of quartz, resulting in high brittleness and a moderate clay mineral content. The average total organic carbon (TOC) content of the Longmaxi Formation shale was determined to be 3% [23,24].

**Table 1.** Analysis of the Longmaxi shale as determined by XRF.

| Component | Proportions (wt%) |
|:---:|:---:|
| Quartz | 58 |
| Chlorite | 6 |
| Plagioclase | 1 |
| Potassium feldspar | 3 |
| Calcite | 10 |
| Dolomite | 2 |
| Hematite | 1 |
| Illite | 19 |

### 2.4. Numerical Simulation Method for Physical Containment

Currently, the understanding of the sealing effects of nanoparticle drilling fluids on shale pore space is primarily based on physical experimental data [25–29]. However, the mechanisms governing the transportation, dynamic stacking, and microscopic sealing of nanoparticles in drilling fluids after entering the shale pore space remain unclear. In this study, we employ fluid dynamics calculations and discrete elements to simulate the microscopic scale sealing of shale pores by particle suspensions. The drag force experienced differs from that of conventional sizes given the nanoscale size of the particles. To ensure accurate fitting and prediction, we compiled experimental data and empirical formulas to develop a customized program to correct the standard drag equation. The schematic diagram depicting the sealing of the shale pores by the nanoparticles is illustrated below (Figure 1).

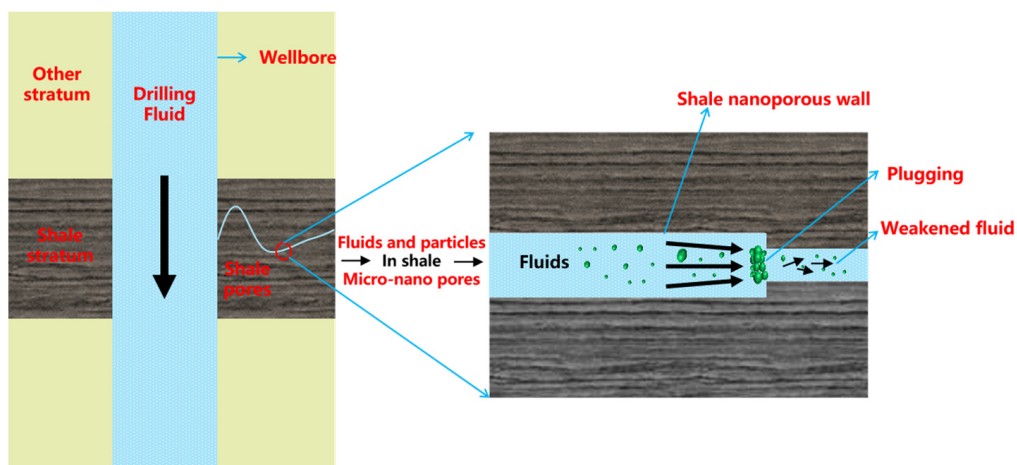

**Figure 1.** Schematic diagram of particles plugging pores.

The particle concentration and the particle size can be adjusted, thus enabling the simulation of shale pore sealing effects under different particle parameters. Effective blocking and bridging cannot be formed with a concentration that is too low. The blocking effect can be quantitatively judged combined with data visualization by monitoring the number of particles and pressure in the pore space [30].

An established particle release area serves as the starting point for particle tracking in the conducted study. Each particle is released at a consistent speed, while their directions are randomized to mimic the real-world process of particle entry into the shale pore space. This approach enables a comprehensive assessment of the entire blocking process, as the transient simulation allows for monitoring the movement of particles at each step. Additionally, the particle size can be adjusted to simulate the combined blocking effect of various particle sizes in later stages. Once the particles are introduced into the shale pore space, their trajectories are tracked through successive particle calculations. A 3D model is used to add a double-precision computational mode, along with a no-slip boundary and an elastic boundary on the shale pores. The model takes into account gravity factors and particle rotation.

Assuming that the drilling fluid is continuous, the nanoparticle drilling fluid can be described by the following equations based on the localized Wiener–Stokes equations according to the mass conservation equation and momentum conservation:

$$\frac{\partial}{\partial t}\left(\rho \vec{v}\right) + \nabla \cdot \left(\rho \vec{v} \vec{v}\right) = -\nabla p + \nabla \cdot (\bar{\bar{\tau}}) + \rho \vec{g} + \vec{F} \tag{1}$$

where $\rho$ is the density (in g/cm$^3$), $S_m$ is the mass of the dispersed phase added to the continuous phase (in kg), $\vec{v}$ is the fluid velocity (in m/s), $p$ is the static pressure (in pa), $\bar{\bar{\tau}}$ is the stress tensor, and $\rho \vec{g}$ and $\vec{F}$ are gravity and external forces (in N), respectively.

For the Reynolds number $R_e$, the formula is:

$$\frac{d\vec{u_p}}{dt} = F_D\left(\vec{u} - \vec{u_p}\right) + \frac{\vec{g}\left(\rho_p - \rho\right)}{\rho_p} + \vec{F} \tag{2}$$

$$R_e = \frac{\rho d_p \left|\vec{u_p} - \vec{u}\right|}{\mu} \tag{3}$$

where $F_D$ is the additional acceleration applied to the particles (in m/s$^2$), $\rho_p$ is the density of the nanoparticles (in g/cm$^3$), $\vec{u_p}$ is the particle velocity (in m/s), $\mu$ is the fluid phase velocity (in m/s), $\vec{F}$ is the drag force per unit particle mass (in N), $\vec{u}$ is the fluid phase viscosity (in mPa·s), and $d_p$ is the nanoparticle diameter (in m).

The equations of particle motion are realized by integrating them stepwise over discrete time steps. The particle trajectory and particle velocity can be calculated using the following equations:

$$\frac{du_p}{dt} = a + \frac{1}{\tau_p}\left(u - u_p\right) \tag{4}$$

where $a$ is the acceleration composed of factors other than the drag force on the particles (in m/s$^2$).

The final nanoparticle new position velocity equation is:

$$u_p^{n+1} = u^n + e^{-\frac{\Delta t}{\tau_p}}\left(u_p^n - u^n\right) - a\tau_p\left(e^{-\frac{\Delta t}{\tau_p}} - 1\right) \tag{5}$$

where $u_p^n$ and $u^n$ are the particle and fluid velocities at moment $n$ (m/s).

When the trapezoidal discretization is applied to the velocity and Reynolds number equations, one obtains:

$$\frac{u_p^{n+1} - u_p^n}{\Delta t} = \frac{1}{\tau_p}\left(u^* - u_p^*\right) + a^n \tag{6}$$

The velocity of the mass at the new position $n + 1$ is given by the following equation:

$$x_p^{n+1} = x_p^n + \frac{1}{2}\Delta t\left(u_p^{n+1} + u_p^n\right) \tag{7}$$

The shale pore model was set up as a zigzag tube; a setting more in line with the particle transport law and more relevant to the actual experimental results, as opposed to a straight tube. The particle size can be adjusted, the material is set as $SiO_2$, and a total of 10 particle sizes are selected to promote particle grading. Each time step is 0.004 s, and the exit diameter is 2 μm. The curved part is the pressure cloud of the fluid and particles. The particles will gather at the bends and the exit. At the same time, the small particles will gradually flow out of the tunnel and the large particles will support each other and block the exit.

The average particle size is adjustable and moderate. They will block the pores directly and will not produce any build-up effect if the particle size is too large. On the contrary, it will be difficult to block the outlet, which leads to long calculation times if the particle size is too small (Table 2).

**Table 2.** Structural parameters of pores and particles.

| Parameters | Numerical Value |
|---|---|
| Pore length, μm | 20 |
| Pore bending, ° | 50 |
| Pore outlet diameter, μm | 2 |
| Particle release area, μm$^2$ | 2 |
| Types of particles of different sizes | 15 |
| Average particle diameter, nm | 300 |
| Compute pattern | Pressure, transient |
| Particle track mode | Nonstationary tracking |
| Discrete phase reflection coefficient | 0.5 |
| Spring cushioning parameters | 1000 |
| Wall surface of grain | Reflection pattern |

The hole is set as a fixed wall with a non-slip interface. Particle collisions are reflective, and the wall reflection coefficients are categorized into normal and tangential recovery coefficients. The discrete phase's reflection coefficient was set to 0.5, considering the silicon dioxide particles' elasticity. The normal contact force between particle collisions was determined based on the Spring–Dashpot model, while the tangential contact force relied on the coefficients of adhesive friction and sliding friction (Table 3). The meshes of the model are categorized into two types, namely structured and unstructured meshes. There are almost no particles available to collide in the early stage, and the mesh can be a coarse mesh. However, the point is on the particle filling process and the mesh in this region must be dense. We selected a fine mesh, considering account time and accuracy considerations. The tetrahedral mesh serves as the primary body mesh and is primarily used for fluid flow and particle migration. The wedge mesh is utilized as the boundary mesh to enhance the accuracy of distinguishing between contact and collision in the boundary layers. The data became more stable as the mesh underwent gradual refinement through the validation of grid independence.

**Table 3.** Fluid physical parameters.

| Physical Property | Parameters |
|---|---|
| Fluid properties | Deionized water |
| Fluid density, kg/m$^3$ | 1000 |
| Fluid viscosity, mPa·s | 1, 3, 5 |
| Temperature, K | 298 |
| Granulation (sugar, chemical product) | Silicon dioxide (SiO$_2$) |
| Particle density, g/cm$^3$ | 2.2 |
| Diffusion coefficient | 4 |
| Particle size distribution | Rosin–Rammler |

*2.5. Experimental Methods of Chemical Inhibition*

The salinity of various salt solutions was assessed using the Novasina Labswift water activity meter, and the results are shown in Figure 2.

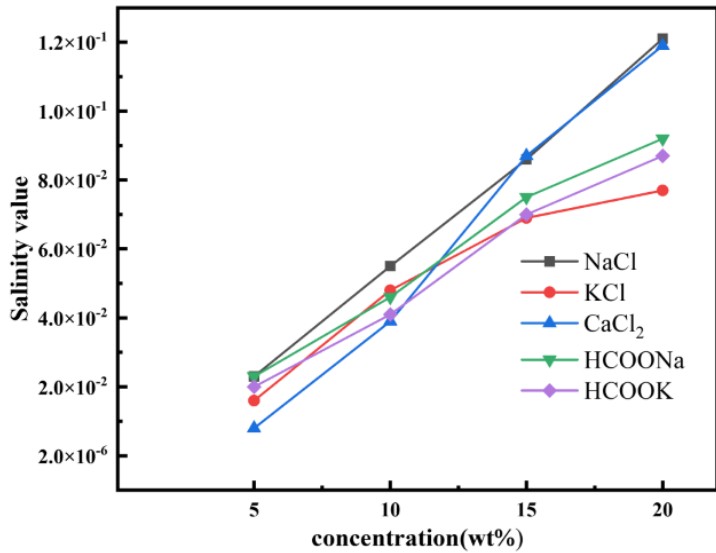

**Figure 2.** Salinity of different types and concentrations of 5–20% salt solutions.

The salinity values of the solutions rose with increasing concentrations of salt solutions. Among the five experimental groups, the NaCl salt solution exhibited the highest salinity value. In contrast, the CaCl$_2$ salt solution displayed the lowest salinity value, with values of 0.008 and 0.039 at concentrations of 5% and 10%, respectively. However, the salinity value of the CaCl$_2$ salt solution experienced a significant increase, reaching 0.119 at a concentration of 20%.

The principle of the pressure transmission experiment is to establish a pressure difference between the upstream and downstream regions of the shale, to maintain constant upstream pressure, and to simultaneously monitor changes in downstream fluid pressure. This allows for an understanding of the pressure transfer process and real-time measurement of fluid permeability. Fluid injection channels and highly sensitive manometers are positioned on the left, right, top, and bottom of the reactor to test the sample permeability simultaneously. Displacement sensors are employed to monitor displacement changes at each load stage in real time. The instrument features a triple parallel vessel kettle at the back to ensure uninterrupted experiments and data monitoring when changing the test solution. The unit is temperature-controlled (rated at 150 °C). Perimeter pressure, upstream pressure, and downstream pressure were set to 2 MPa, 1.5 MPa, and 0 MPa, respectively. The experimental patterns of pressure transfer to shale from five types and three concentrations of salt solutions were tested using the PTE pressure transfer device. Experimental data were recorded every 1 min [31,32].

## 3. Results

### 3.1. Results of Chemical Inhibition Experiments

Figure 3 presents the pressure transmission experimental data for 5 wt% $CaCl_2$ [33]. The data show that the downstream pressure did not increase significantly for 23 h, while the upstream pressure was maintained at a constant level. The initial permeability was high relative to the 5 wt% NaCl and 5 wt% KCl solutions [34], and the initial permeability was higher. The time point when the permeability started to increase significantly was 33 h. The permeability increased from $2.0 \times 10^{-4}$ mD to $1.8 \times 10^{-3}$ mD in 11.11 h. When the downstream pressure began to increase, the permeability gradually increased to $9 \times 10^{-3}$ mD after 1 h, when the downstream pressure was consistent with the upstream pressure. The time required to open the pore channel and the time required to increase the downstream pressure was shorter relative to the 5 wt% NaCl solution [35].

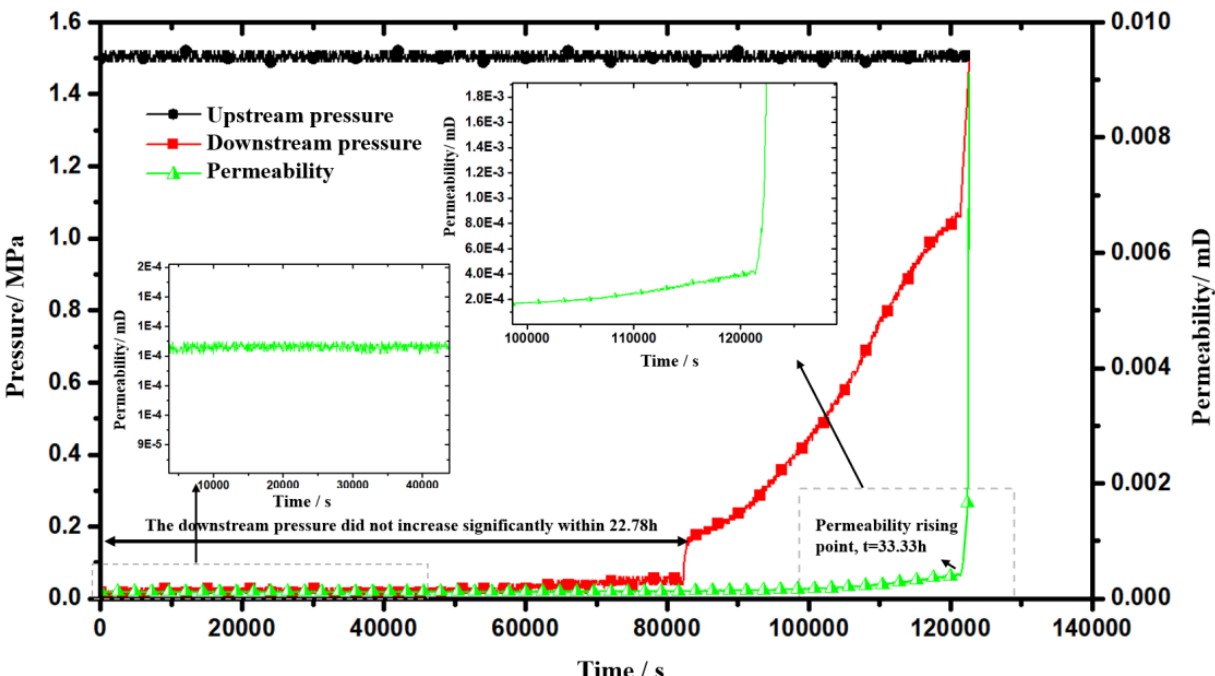

**Figure 3.** Plot of experimental data for pressure transmission of 5 wt% $CaCl_2$.

The experimental data indicate that a 10% concentration of salt solution is not effective in inhibiting shale pressure transfer [36]. Furthermore, shale pressure transfer experiments using various shale types can help elucidate the impact of salt solutions on the stabilization of the shale wellbore. Determining the precise law governing the maintenance of wellbore stability under salt solutions is challenging due to numerous factors influencing pore pressure transfer. This is because the maintenance of the wellbore stabilization law by salt solutions containing different salt ion types is not easy to determine. These include salt ion concentration, shale clay mineral composition, and salt chemical molecular structure.

The inhibition effect of $Ca^{2+}$ is stronger when the montmorillonite content is higher and the salt solution concentration is lower [22]. However, the inhibition effect of $Ca^{2+}$ ions significantly decreases, and the inhibition effect of $K^+$ ions becomes more pronounced when the salt concentration is raised to 10% or even 20% [37]. This pattern aligns with experimental results demonstrating that salt solutions block pressure transfer at low concentrations [38].

Summarizing the results of the Longmaxi Formation shale pressure transmission experiments, the optimal type and concentration of salt solution are 20 wt% HCOONa (Figure 4). Therefore, it is not the case that the higher the concentration of the salt solution, the more effective it is in blocking the shale pressure transmission.

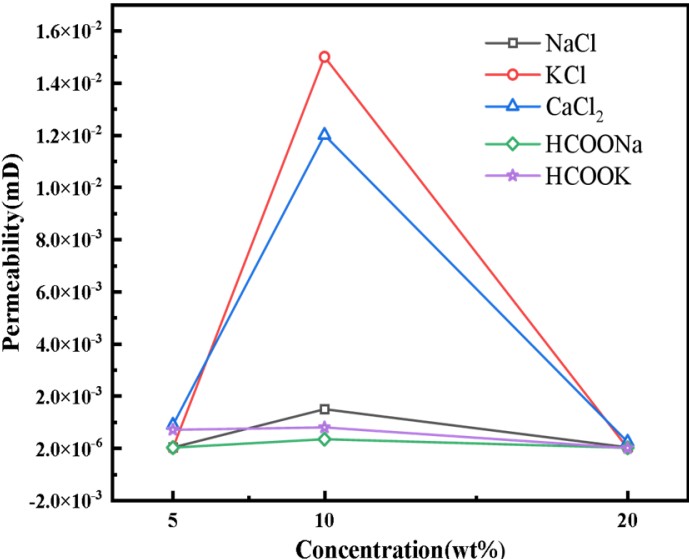

**Figure 4.** Summary of Longmaxi shale seepage process results.

### 3.2. Numerical Simulation Results of Physical Blocking

The particles gradually accumulate in the bent pore space with increasing transport times, as observed in Figure 5 [39]. The particles will collide and rebound during the transportation process, and the particles mainly accumulate in the pore bends and are gradually transported to the pore exit [40]. The particles are of different scales, and the fluid–solid coupling model ensures that the ratio of the average diameter of the particles entering the pore space to the pore exit diameter is fixed. The real-time velocity, displacement, rotation, and other flow characteristic parameters of the particles transported in the pore space can be recognized and recorded by the model [41]. Characteristic parameters of particle transport, such as particle-to-particle, particle-to-fluid, and particle-to-wall collision, and rebound, can be monitored and recorded [40].

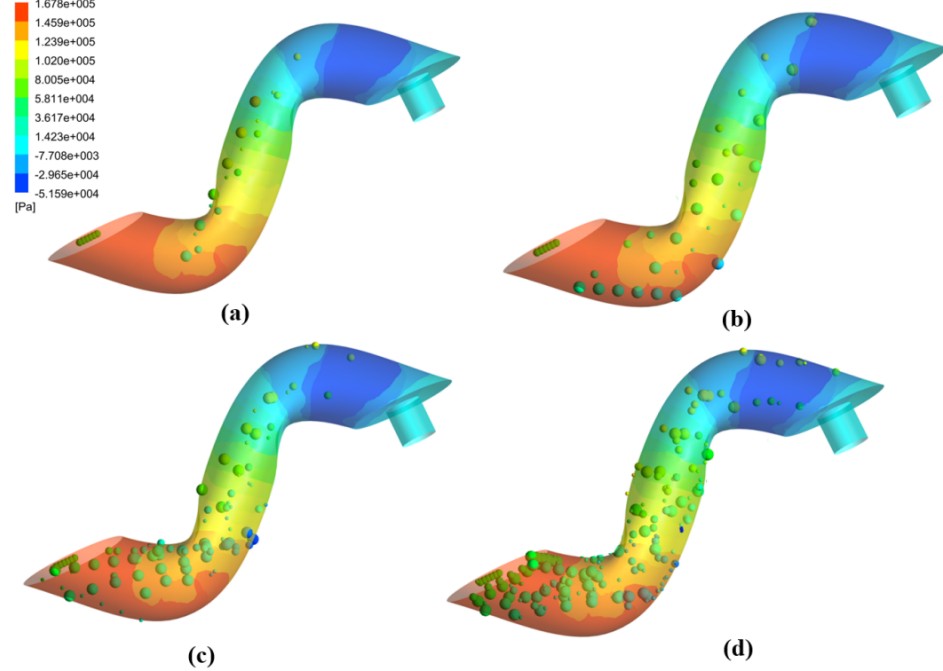

**Figure 5.** Transient maps of particle-blocked pores at different time steps. (**a–d**) mean the transient figures at different time steps, the time steps are 10, 20, 30 and 40 respectively

## 4. Discussion

The effects of particle size, particle concentration, particle shape, particle density, and fluid viscosity on the plugging effect based on the above discrete element model were discussed and analyzed. The first factor to consider is the influence of nanoparticle size on pore sealing effectiveness. We have set particle sizes at 1/2, 1/3, and 1/5 of the pore exit size. It is important to note that the released particle size should not exceed the pore exit size. The single particle would block the exit entirely, preventing the stacking and filling process and making it impossible to discern how other particle parameters affect the blocking process if the size of the released particles exceeds the pore outlet size. As depicted in Figure 6, the number of stacked particles increased and gradually stabilized over time when the particle concentration was 5%. Notably, the effect of particle stacking for particle size 1/2 increased by 13% and 23% relative to particle sizes 1/3 and 1/5, respectively.

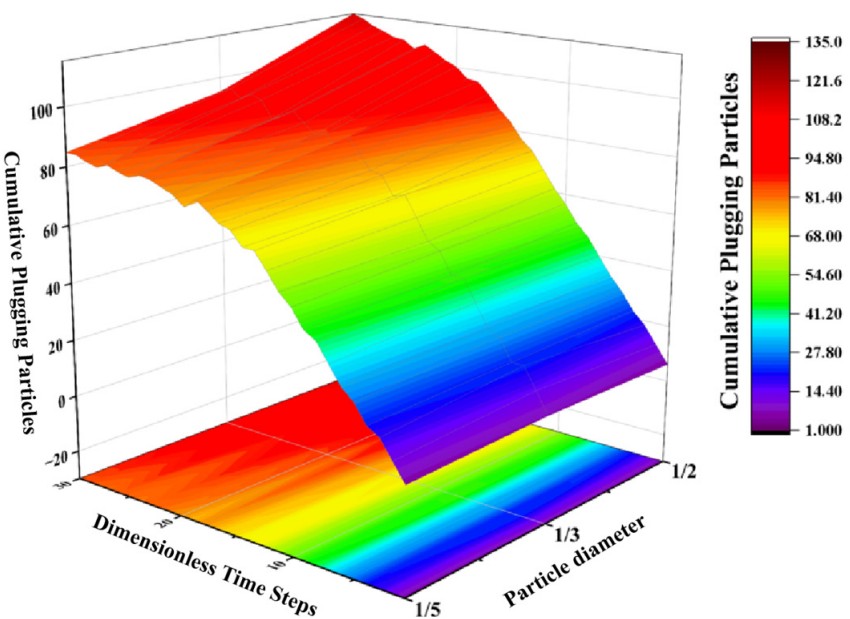

**Figure 6.** Effects of particle diameter on plugging efficiency.

Nanoparticle concentration also plays a significant role in pore sealing. The higher the concentration of nanoparticles in the same amount of time, the higher the probability and number of particles establishing bridges. Additionally, low nanoparticle concentrations require more time to establish effective bridges or may fail to create effective bridges altogether. Conversely, the number of particles required to seal the pores will be reduced accordingly, and a rapid blocking process might indirectly diminish the impact of nanoparticle concentration on sealing effectiveness if the particle size is set too large and the pore size remains unchanged. The results demonstrate that the blocking efficiency increased by 75% and 50%, respectively, at 11 wt% and 5 wt% particle concentrations compared to the 1 wt% particle concentration (Figure 7).

The contact force relationship between the particles is more complex when the particles are not spherical. The DEM model can be mathematically approximated by non-circular particles so that they can be calculated as circular particles. Particle collision models can be used to model the contact of circular particles based on this approach.

At this point, the shape factor parameter is needed to make a mathematical approximation, $\theta$, defined as:

$$\theta = \frac{s}{S} \tag{8}$$

where $s$ is the surface area of the sphere and $S$ is the actual surface area of the particle (in cm$^2$).

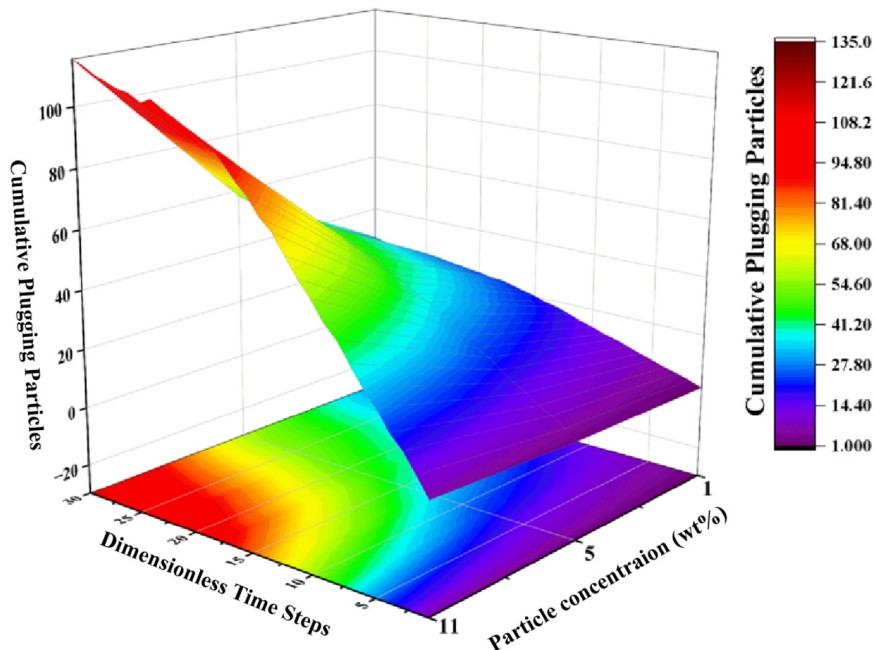

**Figure 7.** Effects of particle concentration on plugging efficiency.

The closer $\theta$ is to 1, the more nanoparticles tend to be spherical. The more irregular the particle shape, the higher the efficiency of the nanoparticles in sealing the shale pores (Figure 8). The sealing efficiency is 14% higher than the sealing efficiency relative to round particles when $\theta$ is 0.25.

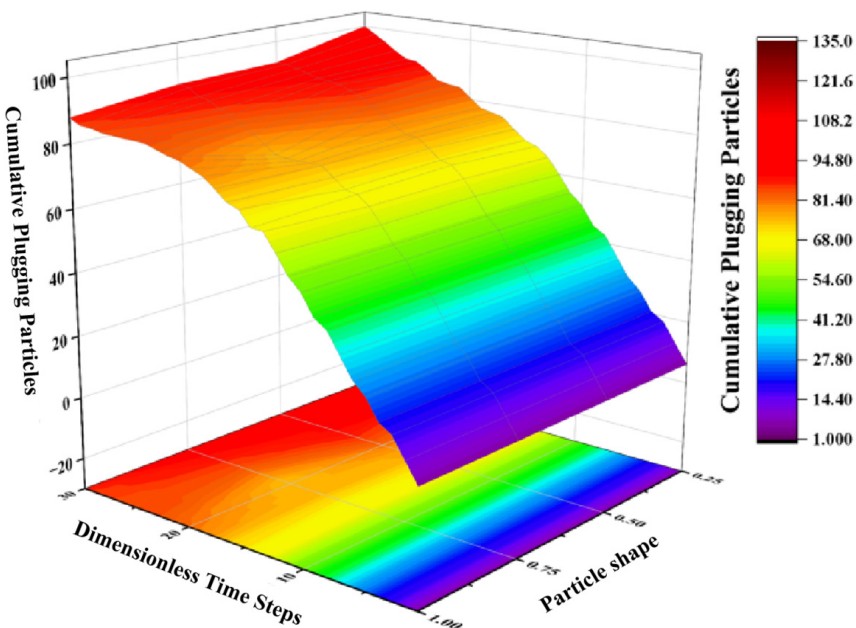

**Figure 8.** Effects of particle shape on plugging efficiency.

Changing the physical properties of nanoparticles can regulate the blocking effects of pore particles. Meanwhile, the effects of changing fluid physical properties on shale pore plugging due to the relationship between particle density and gravity are investigated. The different particle densities affect the distribution of the particles in the pores. Under the setting conditions, it is not the case that the higher the particle density is, the higher the plugging efficiency is (Figure 9). One reason is that the particles are so heavy that they mostly concentrate in the lower part of the throat, which is not conducive to blocking.

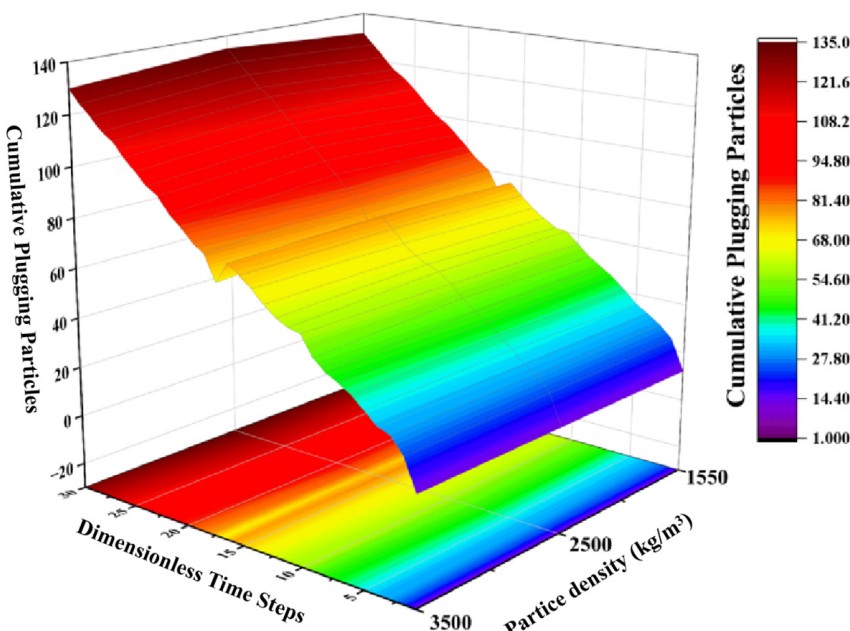

**Figure 9.** Plugging effects of different particles densities.

Figure 10 illustrates the nanoparticle blocking efficiency at various viscosities for a nanoparticle concentration of 1 wt% [42]. The experimental findings indicate a decrease in the accumulation of particles as the calculation time increases. This hinders the formation of an effective blockage, which can primarily be attributed to the low concentration of particles. Consequently, the outlet remains inadequately sealed, causing particles to flow out of the fluid during the initial stage of accumulation. However, an increase in viscosity significantly enhances the sealing effect when the nanoparticle concentration is 1 wt%. Specifically, the sealing efficiency of a 5 mPa·s nanoparticle solution surpasses that of a 1 mPa·s nanoparticle solution by 16% (Figure 10). Therefore, elevating the viscosity to 5 mPa·s proves to be an effective approach for improving the blocking efficiency.

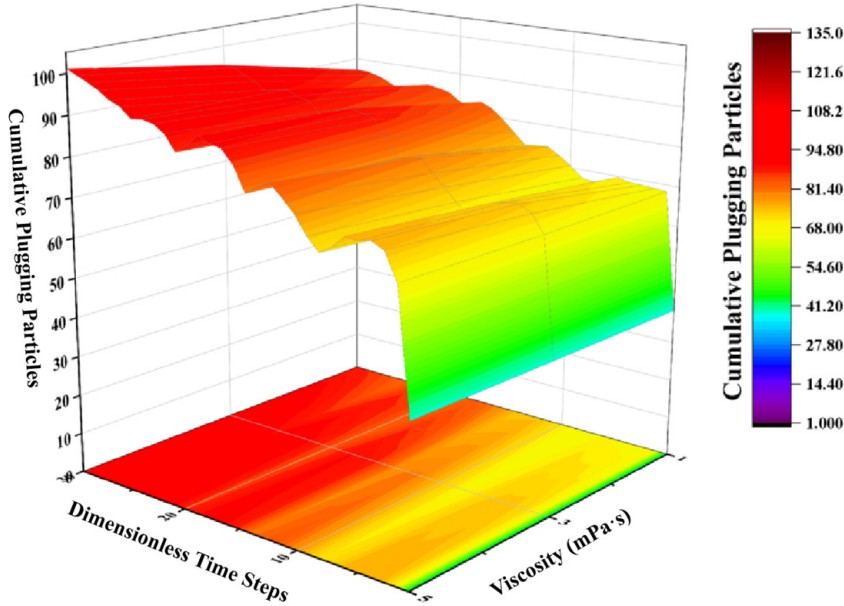

**Figure 10.** Comparison of plugging effects of nanoparticle solutions with different viscosities.

### 5. Conclusions

In this paper, a pressure transfer experimental platform was used to study the influence the laws of different hydroactive salt solutions and their effects on shale permeability and borehole stability. At the same time, numerical simulations based on a discrete element hydrodynamic model were used to reconstruct the force model of particles in pore spaces and to study the blocking law of particle parameters and fluid physical properties and their effects on shale pore space.

1.  The salt solution is most capable of blocking the pressure transfer in the shale pore space when the salt solution adopts a formate system with a water activity of 0.092. At this time, the drilling fluid does not easily penetrate into the shale pore space and is more capable of maintaining the shale wellbore stability.
2.  Nanoparticle concentration is the most critical factor affecting shale pore plugging efficiency for physical plugging numerical simulations. Particle size has a large effect on the blocking efficiency of shale pores, and a 30% increase in particle diameter increases the pore blocking efficiency by 13% when the maximum particle size is smaller than the pore outlet.
3.  Particle density has a small effect on the final sealing effect of pores. The pore sealing efficiency was only increased by 4%, and the particle density was increased by 60%.
4.  Fluid viscosity is significant for shale pore sealing. The sealing efficiency of the particles in the pore space is increased by approximately 16% when the fluid viscosity is increased to 5 mPa·s.

These findings provide an important theoretical and technical basis for the selection of water-based drilling fluid systems for long horizontal drilling of shale gas, especially for improving shale oil and gas production. Through a combination of experiments and numerical simulations, this study has made innovative progress in understanding and controlling the interactions between shale formations and drilling fluids.

**Author Contributions:** Conceptualization, Y.S.; Methodology, W.L.; Software, H.C.; Validation, X.M.; Formal analysis, J.L. and Z.X.; Resources, Y.Z.; Data curation, J.L., H.C. and Y.Z.; Writing—review & editing, Y.S. and H.Z.; Visualization, Z.X.; Supervision, Y.S.; Project administration, W.L. All authors have read and agreed to the published version of the manuscript.

**Funding:** This research was funded by China Geological Survey, Grant Number DD20230025.

**Data Availability Statement:** Data are contained within the article.

**Conflicts of Interest:** The authors declare no conflict of interest.

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
