# Peer review of "Experimental Water Activity Suppression and Numerical Simulation of Shale Pore Blocking"

_processes, doi:10.3390/pr11123366_

Round 1
Reviewer 1 Report
Comments and Suggestions for Authors
The review of the manuscript entitled: “Experimental water activity suppression and numerical simulation of shale pore blocking for maintaining wellbore stability”. The work is very interesting, novel, and has a scientific style. Also, the work is within the journal's scope, and it is easy to follow. By responding to the following comments and questions, the work can be ready for publication:
1. The exact and main findings should be presented in the abstract.
2. Please also add the mechanism for improvement.
3. The type of nanoparticles should be added.
4. The gap and novelty should be presented clearly.
5. It is recommended to mention the other technologies for oil production improvement such as asphaltene control in the introduction section. For this purpose, the next references can be used in the revision stage:
https://doi.org/10.1016/B978-0-323-90510-7.00004-5
https://doi.org/10.1080/10916466.2022.2049819
6. It is recommended to add the rock properties of the used core samples.
7. The units of the variables used in the equations should be provided.
8. Why was the deionized water used? It cannot simulate the reservoir condition. It is a limitation for this paper.
9. In fig. 2, wt.% in what? Also, the salinity value does not have a unit?
Author Response
Please check the attachment for point-to-point revision.
Response to Reviewer â… Comments |
||
1. Summary |
|
|
Thank you very much for taking the time to review manuscript. The introduction and background have been improved. The cited references were updated. The methods were adequately described and the results were modified and improved. Please find the detailed responses below and the corresponding revisions.
|
||
2. Point-by-point response to Comments and Suggestions for Authors |
||
Comments 1: The exact and main findings should be presented in the abstract。 Please also add the mechanism for improvement. |
||
Response 1: Thank you for the suggestion to add and improve the content of the article. The key findings have been numbered in the summary. The revised contents were reflected in lines 13-26.
|
||
Comments 2: Please also add the mechanism for improvement. |
||
Response 2: Thank you for the suggestion. Narrative on the mechanism of influence of factors affecting pore sealing efficiency were added. The revised contents were marked with yellow in lines 212-215; 322-329
|
||
Comments 3: The type of nanoparticles should be added. |
||
Response 3: Thank for your suggestion. The type of nanoparticles in this article is silica nanoparticles, which are elastic and chemically stable. It is explained in the original manuscript.
|
||
Comments 4: The gap and novelty should be presented clearly. |
||
Response 4: Thank you for your suggestions to add and improve the article. The article has been supplemented and improved. The innovation of the article is clarified by changing the content of lines 114-122.
|
||
Comments 5: It is recommended that references to other oil enhancement techniques such as asphaltene control be included in the introductory section. For this purpose, the following references could be used at the revision stage: https://doi.org/10.1016/B978-0-323-90510-7.00004-5 https://doi.org/10.1080/10916466.2022.2049819
|
||
Response 5: Thank you for your suggestions, reference research was conducted on other oil enhancement techniques such as asphaltene control and cited in lines 401-404. And cited references are highlighted in yellow. References: Ghamartale, A.; S. Afzali; N. Rezaei; S. Zendehboudi. "Chapter Three - Fundamentals of Chemical Inhibitors of Asphaltenes." In Asphaltene Deposition Control by Chemical Inhibitors, edited by Ali Ghamartale, Shokufe Afzali, Nima Rezaei and Sohrab Zendehboudi, 47-83: Gulf Professional Publishing, 2021. Khormali, A. "Effect of Water Cut on the Performance of an Asphaltene Inhibitor Package: Experimental and Modeling Analysis." Petroleum Science and Technology 40 (2022): 2890-2906.
|
||
Comments 6: It is recommended to add the rock properties of the used core samples. |
||
Response 6: Thank you for the suggestion to supplement the compositional analysis of the Longmaxi Formation rock samples used in the experiment. The above content is mainly concentrated in lines 146-151 of the manuscript.
|
||
Comments 7: The units of the variables used in the equations should be provided. |
||
Response 7: Thank you for your suggestions. The units of variables appearing in the paper have been supplemented, mainly focusing on lines 186-200 of the article.
|
||
Comments 8: Why was the deionized water used? It cannot simulate the reservoir condition. It is a limitation for this paper. |
||
Response 8: Thank you for the suggestion. The deionized water is pure water with impurities removed in ionic form, which would better represent the effect of different concentrations of salt solutions on fluid water activity. The above content is explained and illustrated in the manuscript.
|
||
Comments 9: In fig. 2, wt.% in what? Also, the salinity value does not have a unit? |
||
Response 9: Thank you for your advice. Salinity itself has no units. It is differentiated by pure water as 1. The value gradually approaches 0 as the salinity increases.
|
||
3. Response to Comments on the Quality of English Language |
||
Point: None. |
||
Response: This editorial expert has no comments in this regard. The language of the article was revised and improved by native speakers. Thank you for your suggestion and time. |

Reviewer 2 Report
Comments and Suggestions for Authors
SUMMARY
The article submitted for review is relevant. The issue of experimental water activity suppression and numerical simulation of shale pore blocking for maintaining wellbore stability is considered. The relevance of the study is due to the fact that shale gas is an extremely important and valuable natural resource. The problem raised in the article has scientific and industrial genesis and is due to the lack of the most effective technologies that ensure stable production of shale gas at a new level and the prevalence of more standard technologies known for a long time. The authors proposed an experimental platform that is used to systematically study the law on the influence of saline solutions of varying water activity on shale permeability and well stability. That is, the authors presented a study that has scientific novelty and practical significance. The results of this study provide a theoretical technical basis for selecting a water-based drilling fluid system for long-distance horizontal shale gas drilling. Considering the scientific novelty of the study, as well as the practical significance, the reviewer believes that the research was carried out at a high level and deserves support. At the same time, there are a number of comments that should be corrected, they are listed below.
COMMENTS
1. The authors gave the title of the article somewhat ponderously; perhaps it should be shortened a little, indicating the main idea proposed in the article and the problem being solved.
2. The beginning of the abstract is noteworthy, namely that there are no formulations of scientific and applied problems. The authors spoke about the relevance of shale gas and that this technology is widespread, but they did not reflect the formulation of scientific and industrial problems. This should be added.
3. The keywords contain general terms that cannot reflect the uniqueness and individuality of the research conducted. Five words and phrases of a general nature are presented, from which it is difficult to determine the clear focus of the article. Authors should increase the number of keywords and make them more relevant to the subject of the study.
4. The Introduction section is very superficial, and the small number of analyzed works attracts attention, namely 11. That is, the authors provided an insufficiently detailed overview of the current state of the issue. A lot of research is devoted to such a topic as drilling and shale gas, both in terms of technology and in terms of modeling, methodology and materials science. The literature review should be reflected by grouping it according to various criteria according to the focus of previously completed articles. At the same time, it is important to reflect the latest research over the past 5 years, since technologies in the field of shale gas have stepped far forward. This will help make the article more relevant to today.
5. Authors should more clearly articulate the research problem and therefore its goal and tasks at the end of the Introduction section.
6. Section 2 looks very uninformative. Such a section does not meet the requirements of the journal. The authors should give a higher degree of attention to the description of materials and tools, but their section 2 occupies only 11 lines from 109 to 119. Such a description of the methodological part is not an appropriate requirement of the journal.
7. The photograph presented in Figure 1 is of low quality, and some images are poorly distinguishable; also, the captions to the figure, written in red font, are sloppy. This figure should be done in higher quality. The same remark applies to Figure 2, as well as Figures 3 and 4.
8. The reviewer notes a drawback of Figure 2, which is that the areas between the points, namely 5, 10 and 20%, are made in the form of straight segments. At the same time, the article does not describe in sufficient detail the nature of the change in the relationship between these percentages. The authors also provide points at 5, 10 and 20%. However, it seems that the 15% mark is missing from the graphical dependencies, although it certainly suggests itself in order to also appear on this graph. Approximately the same remark applies to Figure 4.
9. As for Figure 3, there are indistinguishable, unreadable symbols on it. It is unacceptable.
10. Figures 5, 6, 7, 8, 9, 10 seem interesting, but they are poorly discussed and poorly explained in the text. The authors should include a discussion of the results obtained in a separate section. The authors presented Section 4 as a summary of the results and their discussion, but this may prevent the reader from assessing the author's specific contributions to science. The Discussion should be separated into a separate section and a detailed comparison of the results obtained with the results previously obtained by other authors should be presented.
11. On line 320 there is a mention of figure 11, but figure 11 is not in the manuscript. Needs clarification and correction.
12. Throughout the text, the font size of references is slightly smaller than the main font. Needs to be fixed.
13. The list of references, including 32 references, is not large enough to talk about a complete analysis of the current state of the issue. The reviewer recommends considering at least 10-15 more works.
14. The reviewer's general opinion of the article is that it is necessary to correct the comments made and make some adjustments to the presentation style and English language.
General conclusion: major revision.

Comments on the Quality of English LanguageMinor edits and correction of English required.
Author Response
Please check the attachment for the revision file.
1. Summary |
|
|
Thank you very much for taking the time to review this manuscript. The article title, abstract and keywords have been optimized. The content and form of expression are refined to reflect the content of the research. Depth and number of references were supplemented. The figures were redrawn and revised regarding the comments that the legend was unclear. The introduction and background have been improved. The cited references were updated. The methods were adequately described and the results were modified. Please find the detailed responses below and the corresponding revisions. |
||
2. Point-by-point response to Comments and Suggestions for Authors |
||
Comments 1: The authors gave the title of the article somewhat ponderously; perhaps it should be shortened a little, indicating the main idea proposed in the article and the problem being solved. |
||
Response 1: Thank you for your suggestion. The title has been streamlined and deepened. More prominence is given to the research methodology - numerical simulation and the main factor affecting pore sealing.
|
||
Comments 2: The beginning of the abstract is noteworthy, namely that there are no formulations of scientific and applied problems. The authors spoke about the relevance of shale gas and that this technology is widespread, but they did not reflect the formulation of scientific and industrial problems. This should be added. |
||
Response 2: Thank you for your suggestions. The beginning of the abstract has been revised and improved to better reflect the research. The revised contents were marked with yellow in lines 4-12.
|
||
Comments 3: The keywords contain general terms that cannot reflect the uniqueness and individuality of the research conducted. Five words and phrases of a general nature are presented, from which it is difficult to determine the clear focus of the article. Authors should increase the number of keywords and make them more relevant to the subject of the study. |
||
Response 3: Thank you for your suggestion. Keywords have been modified. Remove sealing and chemical inhibition, and increase nanoscale pores and salt ion inhibition.
|
||
Comments 4: The Introduction section is very superficial, and the small number of analyzed works attracts attention, namely 11. That is, the authors provided an insufficiently detailed overview of the current state of the issue. A lot of research is devoted to such a topic as drilling and shale gas, both in terms of technology and in terms of modeling, methodology and materials science. The literature review should be reflected by grouping it according to various criteria according to the focus of previously completed articles. At the same time, it is important to reflect the latest research over the past 5 years, since technologies in the field of shale gas have stepped far forward. This will help make the article more relevant to today. |
||
Response 4: Thank you for your suggestions. New content has been added to provide a more complete citation discussion. The revised contents were marked with yellow in lines 401-408; 414-430. Coupled hydrochemistry-hydraulics modeling was added, and the effects of salt dissolution has been discussed. Recrystallization on reservoir porosity to make the introduction more reflective of the research on salt for shale plugging and the great technological advances that have been made in the field of shale gas.
|
||
Comments 5: Authors should more clearly articulate the research problem and therefore its goal and tasks at the end of the Introduction section. |
||
Response 5: Thank you for your suggestions. The introduction has been optimized to more clearly state the research questions and objectives. The research questions and conclusions are added at the end of the introduction. It makes the article structure more reasonable. The revised contents were marked with yellow in lines 123-132 of the paper.
|
||
Comments 6: Section 2 looks very uninformative. Such a section does not meet the requirements of the journal. The authors should give a higher degree of attention to the description of materials and tools, but their section 2 occupies only 11 lines from 109 to 119. Such a description of the methodological part is not an appropriate requirement of the journal. |
||
Response 6: Thank you for your suggestion. The structure of the article has been adjusted, Chapter 2 has been merged into Chapter 3, and shale xrd composition analysis has been added to make each part more substantial. The revised contents were marked with yellow in lines 133-151.
|
||
Comments 7: The photograph presented in Figure 1 is of low quality, and some images are poorly distinguishable; also, the captions to the figure, written in red font, are sloppy. This figure should be done in higher quality. The same remark applies to Figure 2, as well as Figures 3 and 4. |
||
Response 7: Thank you for your suggestion. This section has been optimized in this article. Figure 1 has been redrawn to improve its clarity. The red font is used to distinguish it from the layers and make it easier for readers to read. Figures 2 and 4 are drawn again using origin, and the images are enlarged to make it easier to read. Figure 3 is not able to show the markers completely due to the large amount of data. Therefore, the graph markers are shown every 20 points of the curve markers.
|
||
Comments 8: The reviewer notes a drawback of Figure 2, which is that the areas between the points, namely 5, 10 and 20%, are made in the form of straight segments. At the same time, the article does not describe in sufficient detail the nature of the change in the relationship between these percentages. The authors also provide points at 5, 10 and 20%. However, it seems that the 15% mark is missing from the graphical dependencies, although it certainly suggests itself in order to also appear on this graph. Approximately the same remark applies to Figure 4. |
||
Response 8: Thank you for your suggestion. The data in Figure 2 are supplemented. Regarding the lack of data in Figure 4, the salt concentrations used are 5%, 10% and the highest is only 20% for the shale water-based drilling fluid system. Therefore, these three concentrations were previously set for permeability testing. Apologies for the oversight this created.
|
||
Comments 9: As for Figure 3, there are indistinguishable, unreadable symbols on it. It is unacceptable. |
||
Response 9: Thank you for the suggestion. Figure clarity has been improved. However, the symbols cannot be fully displayed due to the large amount of data. Therefore, the notation in the figure displays a curve mark every 20 points.
|
||
Comments 10: Figures 5, 6, 7, 8, 9, 10 seem interesting, but they are poorly discussed and poorly explained in the text. The authors should include a discussion of the results obtained in a separate section. The authors presented Section 4 as a summary of the results and their discussion, but this may prevent the reader from assessing the author's specific contributions to science. The Discussion should be separated into a separate section and a detailed comparison of the results obtained with the results previously obtained by other authors should be presented. |
||
Response 10: Thank you for your advice. Results and discussion sections are separated. The simulated cloud image results are included in the results section. The effects of particle size, particle concentration, particle shape, particle density and fluid viscosity on plugging were discussed, compared and analyzed. The modified content in the original text has been marked in yellow. The revised content is shown on lines 300-303.
|
||
Comments 11: On line 320 there is a mention of figure 11, but figure 11 is not in the manuscript. Needs clarification and correction. |
||
Response 11: Thank you for your suggestion. Such an error has been noted and the article corrected accordingly. Figure 11 is changed to Figure 10. The revised content is shown on line 358.
|
||
Comments 12: Throughout the text, the font size of references is slightly smaller than the main font. Needs to be fixed. |
||
Response 12: Thank you for your suggestion. The reference font has been corrected with reference to the text font. And the font size has been standardized to size 10.
|
||
Comments 13: The list of references, including 32 references, is not large enough to talk about a complete analysis of the current state of the issue. The reviewer recommends considering at least 10-15 more works. |
||
Response 13: Thank you for your suggestions. New content has been added to provide a more complete citation discussion. The revised content is shown on lines 401-408; 414-430.
|
||
Comments 14: The reviewer's general opinion of the article is that it is necessary to correct the comments made and make some adjustments to the presentation style and English language. |
||
Response 14: Thank you for your suggestion. The language of the article was revised and improved by native speakers. |
||
Comments 15: Add this essay citation. |
||
Response 15: Thank you for the suggestion to add some literatures under the redirection in references. References: 1. Ghamartale, A.; S. Afzali; N. Rezaei; S. Zendehboudi. "Chapter Three - Fundamentals of Chemical Inhibitors of Asphaltenes." In Asphaltene Deposition Control by Chemical Inhibitors, edited by Ali Ghamartale, Shokufe Afzali, Nima Rezaei and Sohrab Zendehboudi, 47-83: Gulf Professional Publishing, 2021. 2. Khormali, A. "Effect of Water Cut on the Performance of an Asphaltene Inhibitor Package: Experimental and Modeling Analysis." Petroleum Science and Technology 40 (2022): 2890-2906. 3. Stel’makh S A, Shcherban’ E M, Beskopylny A N, et al. "Influence of recipe factors on the structure and properties of non-autoclaved aerated concrete of increased strength". Applied Sciences, 2022, 12(14): 6984. 4. Medici, G. and West, L.J. "Review of groundwater flow and contaminant transport modelling approaches for the Sherwood Sandstone aquifer, UK; insights from analogous successions worldwide". Quarterly Journal of Engineering Geology and Hydrogeology, (2022), 55(4): qjegh2021-176. 5. Fisher, Q., Kaminskaite, I. and del Pino Sanchez, A. Shale barrier performance in petroleum systems: implications for CO2 storage and nuclear waste disposal. Geoenergy, (2023): 2023-006. 6. Ruan, J.Y.; M.J. Lu; W. Zhang; Y.X. Zhang; Y.H. Zhou; J. Gong; F. Wang; Y.X. Guan. "Optimization of the Plugging Agent Dosage for High Temperature Salt Profile Control in Heavy Oil Reservoirs." FDMP-FLUID DYNAMICS & MATERIALS PROCESSING 19 (2023): 421-436. 7. Zhang, C.; K.H. Lv; J.Q. Gong; Z. Wang; X.B. Huang; J.S. Sun; X.Y. Yao; K.C. Liu; K.S. Rong; M. Li. "Synthesis of a Hydrophobic Quaternary Ammonium Salt as a Shale Inhibitor for Water-Based Drilling Fluids and Determination of the Inhibition Mechanism." JOURNAL OF MOLECULAR LIQUIDS 362 (2022). 8. Fu, L.P.; K.L. Liao; J.J. Ge; Y.F. He; W.Q. Huang; E.D. Du. "Preparation and Inhibition Mechanism of Bis-Quaternary Ammonium Salt as Shale Inhibitor Used in Shale Hydrocarbon Production." JOURNAL OF MOLECULAR LIQUIDS 309 (2020). 9. Lago, F.R.; J.P. Gonçalves; J. Dweck; A.L.C. da Cunha. "Evaluation of Influence of Salt in the Cement Hydration to Oil Wells." MATERIALS RESEARCH-IBERO-AMERICAN JOURNAL OF MATERIALS 20 (2017): 743-747. 10. Chen, Q.; F. Wang. "Mathematical Modeling and Numerical Simulation of Water-Rock Interaction in Shale under Fracturing-Fluid Flowback Conditions." Water Resources Research 57 (2021): e2020WR029537. 11. Wang, H.; W. Liu. "Research on Numerical Simulation Method of Salt Dissolution and Recrystallization of Inter-Salt Shale Oil Reservoir." Journal of Petroleum Science and Engineering 213 (2022): 110464.
|
||
3. Response to Comments on the Quality of English Language |
||
Point 1: None. |
||
Response 1: This editorial expert has no comments in this regard. The language of the article was revised and improved by native speakers. Thank you for your suggestion and time.
|

Reviewer 3 Report
Comments and Suggestions for Authors
General comments
Valid petrophysical-engineering research that fits well the scope either journal or special issue. However, detail is missing and all the 11 specific points need to be addressed before publication. I am available to review the manuscript a second time under request of the editors.
Specific comments
1. Lines 39-41. “The shale formation is characterized by a significant abundance of micropores, and 39 reactive clay minerals such as kaolinite, saponite, and montmorillonite. There are two 40 main reasons for wellbore destabilization”. Add a sentence where you include scholars that have treated research on intergranular/fracture flow and interactions with minerals that can either close (by precipitation) pathways or dissolve (by dissolution) by creating new ones. See recent and relevant literature on the topic:
- Medici, G. and West, L.J., 2023. Reply to discussion on ‘Review of groundwater flow and contaminant transport modelling approaches for the Sherwood Sandstone aquifer, UK; insights from analogous successions worldwide’ by Medici and West (QJEGH, 55, qjegh2021-176). Quarterly Journal of Engineering Geology and Hydrogeology, 56(1), pp.qjegh2022-097.
- Fisher, Q., Kaminskaite, I. and del Pino Sanchez, A., 2023. Shale barrier performance in petroleum systems: implications for CO2 storage and nuclear waste disposal. Geoenergy, pp.geoenergy2023-006.
2. Line 45. Can you be more specific when you use the word “salt”? Are you talking about a specific mineral/chemical specie?
3. Line 107. Clearly state the objectives of your research using numbers (e.g., i, ii, and iii)
4. Line 107. The specific objectives should be 4 due to the fact that you conclusions are characterized by 4 points.
5. Line 140. Specify type/nature of particles to help the reader.
6. Line 269. Specify the scale ranges.
7. Lines 333-346. Add an introduction before the four conclusive points.
8. Lines 333-346. Insert a “take ho message” for the researchers working in your field after your four conclusive points.
9. Lines 348-411. Add the 2 relevant and recent papers suggested above on dissolution/precipitation of chemical species in geological porous and fractured media.
Figures and tables
10. Add the spatial scale to the conceptual model in Figure 1.
11. Figures 6-10. Strain on labels on some axes. Fix the issue.
Comments on the Quality of English LanguageMinor edits for the language are recommended
Author Response
Please check the attachment for the revision file.
1. Summary |
|
Thanks very much for taking your time to review this manuscript. The logic of the manuscript has been optimized. The main purpose and conclusion of the paper are numbered. References on fracture flow and interaction with minerals were supplemented regarding the issues of insufficient references and insufficient depth. The introduction and background have been improved. The cited references were updated. The methods were adequately described and the results were modified and improved. Please find the detailed responses below and the corresponding revisions.
|
|
2. Point-by-point response to Comments and Suggestions for Authors |
|
Comments 1: Lines 39-41. “The shale formation is characterized by a significant abundance of micropores, and 39 reactive clay minerals such as kaolinite, saponite, and montmorillonite. There are two 40 main reasons for wellbore destabilization”. Add a sentence where you include scholars that have treated research on intergranular/fracture flow and interactions with minerals that can either close (by precipitation) pathways or dissolve (by dissolution) by creating new ones. See recent and relevant literature on the topic:
- Medici, G. and West, L.J., 2023. Reply to discussion on ‘Review of groundwater flow and contaminant transport modelling approaches for the Sherwood Sandstone aquifer, UK; insights from analogous successions worldwide’ by Medici and West (QJEGH, 55, qjegh2021-176). Quarterly Journal of Engineering Geology and Hydrogeology, 56(1), pp.qjegh2022-097.
- Fisher, Q., Kaminskaite, I. and del Pino Sanchez, A., 2023. Shale barrier performance in petroleum systems: implications for CO2 storage and nuclear waste disposal. Geoenergy, pp.geoenergy2023-006. |
|
Response 1: Thank you for your suggestions. The content on fracture flow and interaction with minerals provided good assistance in writing this article. References are cited about fracture flow and mineral interactions on lines 45-47; 414-419. Reference: 1. Medici, G. and West, L.J. "Review of groundwater flow and contaminant transport modelling approaches for the Sherwood Sandstone aquifer, UK; insights from analogous successions worldwide". Quarterly Journal of Engineering Geology and Hydrogeology, (2022), 55(4): qjegh2021-176. 2. Fisher, Q., Kaminskaite, I. and del Pino Sanchez, A. Shale barrier performance in petroleum systems: implications for CO2 storage and nuclear waste disposal. Geoenergy, (2023): 2023-006.
|
|
Comments 2: Line 45. Can you be more specific when you use the word “salt”? Are you talking about a specific mineral/chemical specie? |
|
Response 2: Thank you for your suggestion. "Salts" are compounds that are soluble in water and produce cations and anions when dissolved, and the salts covered in this article include NaCl, CaCl2, KCl, HCOONa, HCOOK, and others.
|
|
Comments 3: Line 107. Clearly state the objectives of your research using numbers (e.g., i, ii, and iii) |
|
Response 3: Thank you for your suggestions. The research objectives have been optimized. The revised contents were reflected in lines 123-132.
|
|
Comments 4: Line 107. The specific objectives should be 4 due to the fact that you conclusions are characterized by 4 points. |
|
Response 4: Thank you for your suggestions. The research objectives have been optimized. The revised contents were reflected in lines 123-132.
|
|
Comments 5: Line 140. Specify type/nature of particles to help the reader. |
|
Response 5: Thank you for your suggestion. The type of nanoparticles in this article is silica nanoparticles, which are elastic and chemically stable. It is explained in the original manuscript. |
|
Comments 6: Line 269. Specify the scale ranges. |
|
Response 6: Thank you for your suggestion. The range of concentrations 5-20% is directly represented by the data. The revised contents were reflected in line 243.
|
|
Comments 7: Lines 333-346. Add an introduction before the four conclusive points. |
|
Response 7: Thank you for your suggestion. A rough summary of the main work and findings of the paper is placed at the beginning of the conclusion. The revised contents were shown in lines 371-376.
|
|
Comments 8: Lines 333-346. Insert a “take ho message” for the researchers working in your field after your four conclusive points. |
|
Response 8: Thank you for your suggestion. The above information has been added.
|
|
Comments 9: Lines 348-411. Add the 2 relevant and recent papers suggested above on dissolution/precipitation of chemical species in geological porous and fractured media. |
|
Response 9: Thank you for your suggestions. The content on fracture flow and interaction with minerals provided good assistance in writing this article. References are cited about fracture flow and mineral interactions on lines 45-47; 414-419.
|
|
Comments 10: Add the spatial scale to the conceptual model in Figure 1. |
|
Response 10: Thank you for your suggestions. Figure 1 has been revised in the manuscript. Representation of the elements shown in figure 1 in a larger legend.
|
|
Comments 11: Figures 6-10. Strain on labels on some axes. Fix the issue. |
|
Response 11: Thank you for the suggestion. The data graph was re-exported and the axis labels were optimized. All modified content has been marked in yellow in the article.
|
|
3. Response to Comments on the Quality of English Language |
|
Point 1: None. |
|
Response 1: This editorial expert has no comments in this regard. The language of the article was revised and improved by native speakers. Thank you for your suggestion and time. |

Round 2
Reviewer 1 Report
Comments and Suggestions for Authors
The work is ready for publication.
Reviewer 2 Report
Comments and Suggestions for Authors
The authors have thoroughly revised the manuscript in strict accordance with the reviewer's comments. I have no further comments on the manuscript.